# ESD Ideas: Extended net zero simulations are critical for informed decision making

Andrew D. King[1,2], Nerilie J. Abram[1,3], Eduardo Alastrué de Asenjo[4,5], Tilo Ziehn[6]

[1]ARC Centre of Excellence for the Weather of the 21st Century
[2]School of Geography, Earth and Atmospheric Sciences, University of Melbourne, Melbourne, Kulin nations, Australia
[3]Research School of Earth Sciences, The Australian National University, Ngunnawal country, Canberra ACT 2601, Australia
[4]Institute of Oceanography, Center for Earth System Research and Sustainability (CEN), University of Hamburg, Hamburg, Germany
[5]Max Planck Institute for Meteorology, Hamburg, Germany
[6]CSIRO Environment, Aspendale, Kulin nations, Australia

*Correspondence to*: Andrew D. King (andrew.king@unimelb.edu.au)

**Abstract.** Climate changes under net zero emissions will take many centuries to play out, particularly in the Southern Hemisphere and in the ocean and cryosphere. New millennial-length Earth System Model simulations are required to better understand committed changes and their dependence on delays in reaching net zero emissions, especially with respect to local and regional extremes.

**Main text.** Earth's climate is rapidly changing in response to anthropogenic greenhouse gas emissions. Humanity must achieve net zero emissions in the second half of the 21st century to have any hope of meeting the Paris Agreement goal of limiting global warming to well below 2°C. However, some aspects of the climate are changing faster than others; notably, there is a high degree of inertia in the ocean and cryosphere and changes in these systems will continue long after net zero emissions are achieved. As a result, regional and local climates will continue to evolve for many centuries, such as the projected warming of the Southern Ocean and neighbouring land regions as demonstrated by King et al., (2024). Despite the policy relevance of understanding regional and local changes in climate extremes, such as droughts or heatwaves, and the long-term implications of 21st century emissions, surprisingly little is known about longer timescale climate changes under net zero emissions. Large-scale changes in temperature, the carbon cycle, and sea level rise have been examined using Earth System Models of Intermediate Complexity (EMICs; Weber, 2010), but local changes and changes in extremes are poorly understood. This knowledge gap is mainly due to a lack of suitable model experiments, specifically using Earth System Models (ESMs). Unfortunately, the lack of extended ESM-based net zero emissions simulations may be leading to a lack of appreciation of the extent of associated long-term consequences.

Prior to the Paris Agreement and nations setting net zero goals, climate projections were primarily based on scenarios of increasing atmospheric carbon dioxide concentrations over the 21st century to determine associated global warming and

other climate impacts. Long-term responses to net zero emissions were mainly studied using EMICs (e.g. Lowe et al., 2009; Plattner et al., 2008; Zickfeld et al., 2013) which offer the benefit of reduced computational expense and the ability to incorporate processes beyond the capability of most ESMs, such as ice sheet dynamics, but do not allow for local and regional climate changes and extremes to be studied robustly. These early net zero studies based on EMICs pointed to committed changes in surface temperatures and ocean circulation but with a large degree of uncertainty between individual models (e.g. Plattner et al., 2008) which has persisted to the most recent generation of EMICs (MacDougall et al., 2020).

In recent years, new multi-model experiments, including new model intercomparison projects (MIPs), have been designed to better address the needs of policymakers to understand the implications of net zero emissions. Multi-model experiments, especially from the Zero Emissions Commitment Model Intercomparison Project (ZECMIP; Jones et al., 2019), which include both ESMs and EMICs, have been used to reach conclusions about global-average temperature and carbon cycle changes under net zero (Borowiak et al., 2024; MacDougall et al., 2020) as well as some limited regional analysis using the ESMs in ZECMIP (Cassidy et al., 2023, 2025; MacDougall et al., 2022). Those studies have, by necessity, focussed on relatively short timescale changes given the limited length of these simulations. Existing net zero emissions simulations in ZECMIP and those planned in new protocols - flat10MIP (Sanderson et al., 2024a) and TipMIP (Winkelmann et al., 2025) – suggest simulations of 300 years in length should be run.

A significant knowledge gap exists around understanding long-term changes in the climate under net zero and the long-lasting effects of a potential delay in emissions cessation. The large uncertainty in long-term climate changes under net zero was highlighted in a recent review by Palazzo Corner et al., (2023) which included an expert assessment of Earth system changes under net zero of which most were deemed to be "speculative" or "low confidence", particularly on timescales beyond a century.

Despite the current lack of multi-model net zero experiments, there is some understanding from other work, using ESMs, that points to climate changes continuing for centuries even if emissions were to cease. For example, Sigmond et al., (2020) ran simulations of CanESM2 under net zero emissions for 500-600 years and found that global-mean sea level continues to rise and the Atlantic Meridional Overturning Circulation (AMOC) starts to recover from its decline under rapid emissions.

A recent study using ACCESS-ESM-1.5 experiments run for 1000 years with net zero emissions commencing at different times in the 21st century found substantial warming of the oceans and decline of Antarctic sea ice extent over the course of the climate stabilisation simulations (King et al., 2024). Furthermore, these projections suggest that delays in achieving net zero emissions by even just a few decades can result in very different magnitudes of long-term commitments in the following centuries under net zero (Alastrué de Asenjo et al., 2025; King et al., 2024). Figure 1 summarises projected changes between 300 years (when most existing net zero experiments end) and 1000 years after net zero emissions are

achieved. These differences suggest there are potentially substantial climate changes expected under net zero emissions beyond the timescale of existing experiments and that these span global and local scales. It is also evident that changes may

be highly dependent on the timing/global warming level at which emissions cease suggesting there is a complex response of long-term climate system changes as a function of time and cumulative emissions (e.g. Gillett et al., 2011).

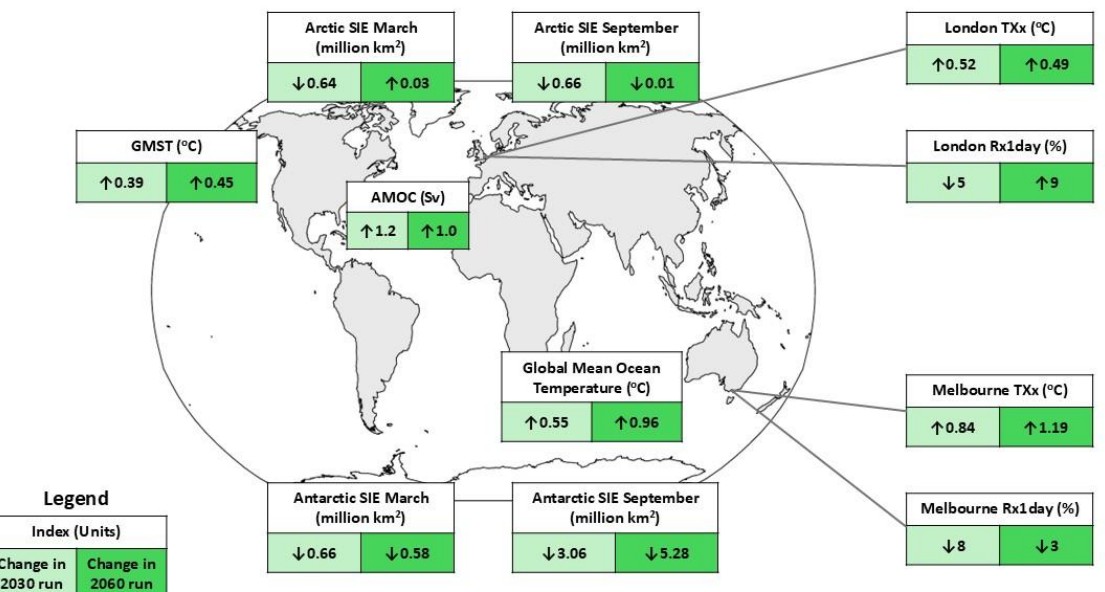

**Figure 1: Summary of changes projected in the climate system between the third century (200-300 years) and the tenth century**
**(900-1000 years) after net zero emissions are imposed in the ACCESS-ESM-1.5 experiments** (King et al., 2024). **The changes in light green cells are from a simulation with emissions cessation in 2030 and the changes in dark green cells are from a simulation where emissions cessation was delayed to 2060. Both of these simulations branch from a high emissions SSP5-8.5 simulation of ACCESS-ESM-1.5. SIE refers to Sea Ice Extent, TXx refers to annual maximum temperature and Rx1day refers to annual maximum 1-day precipitation total. Note, Arctic sea ice extent in September falls to very low levels after net zero emissions is**
**imposed from 2060 and does not recover hence the minimal change identified.**

Policymakers require more regional information for robust decision-making but are not well served by the lack of multi-model net zero experiments on long timescales. This is particularly problematic for understanding changes in interannual-to-multidecadal climate variability for which changes are only likely to be detectable over long periods. The same is true for multi-year extremes, such as multi-year droughts (Falster et al., 2024), for which limited information may be gained from

85 shorter simulations. For example, multi-year droughts in Australia have major impacts and are changing in a warming world (Falster et al., 2024). Analysis of extended ACCESS-ESM-1.5 experiments points to some rainfall recovery (King et al., 2024), but this is only one model and there is high uncertainty even in 21st century projections in this region. Analysis of

multiple extended net zero emissions simulations at different cumulative emissions/global warming levels based on many ESMs would be necessary for policymakers to gain a more robust understanding of future drought hazards in Australia.

The lack of extended net zero simulations risks understating the long-lasting impacts of increased global temperatures relative to pre-industrial levels. However, ESM simulations longer than a few hundred years are computationally expensive, so a compromise needs to be reached. We would suggest that modelling centres run two 1000-year-long simulations as extensions of existing plans. We recommend these could either be at the 1000PgC and 1500PgC cumulative emissions levels (in the flat10MIP framework; Sanderson et al., 2024a) or at the 2°C and 4°C global warming levels (in the TipMIP framework; Winkelmann et al., 2025). This would constitute extensions to existing simulations and would minimise additional computational expense. The flat10MIP and TipMIP frameworks, which make use of emissions-driven model simulations (Sanderson et al., 2024b), are likely to be a part of future phases of the Coupled Model Intercomparison Project (CMIP), so there may be continuity of net zero extensions using future generations of ESMs. It is important to note that some ESMs do not conserve mass or energy sufficiently; therefore, to ensure modelled changes are due to adjustments under net zero emissions, only models with limited drift in piControl simulations (Irving et al., 2020) should be used for such extensions.

We believe such simulations will play a critical role in improving understanding of long-term changes under net zero emissions, including for:

● Quantifying the long-term effect of delay in achieving net zero emissions. The ACCESS-ESM-1.5 analysis suggests changes under net zero emissions will greatly depend on whether emissions cessation is delayed (King et al., 2024). A multi-model analysis of delay in achieving net zero (with higher cumulative emissions levels and peak global warming) is vital to examine the implications for the climate over the coming centuries. For example, analysis of the consequences of delay in achieving net zero emissions for regional droughts is necessary for informing adaptation policy (e.g. the need for water supply augmentation; Henley et al., 2019).

● Identification and constraint of differences in model responses for uncertain changes. Analysis of ZECMIP suggests that local and even global changes under net zero emissions are highly uncertain (Borowiak et al., 2024; Cassidy et al., 2023; MacDougall et al., 2022). At present, uncertainties in long-term changes are unknown.

● Robust quantification of mean and extreme climate changes under net zero emissions. Longer simulations should aid in better estimation of local changes and changes in high-impact extremes, such as drought.

● Identification of potential for tipping points in the climate under different cumulative emissions/global warming levels under net zero emissions. Analysis of the ACCESS-ESM-1.5 millennial length simulations suggests Antarctic sea ice extent will shrink for many centuries (King et al., 2024). Analysis of changes in Antarctica and

other highly vulnerable aspects of the Earth system using multiple models would help constrain these projected changes.

● Analysis of changes in climate variability. Subtle changes in ENSO and some recovery in AMOC have been identified in existing analyses (King et al., 2024; MacDougall et al., 2022; Sigmond et al., 2020), but these are likely to be model-dependent results. Changes in ENSO and AMOC, among other climate modes, must be better quantified for understanding long-term regional climate changes through their teleconnections.

In our view, the set-up and analysis of such extended net zero simulations ahead of the Seventh Assessment Report of the Intergovernmental Panel on Climate Change is necessary and we hope that this suggestion is given due consideration.

**Data Availability**

The data used in this analysis is available here: 10.5281/zenodo.13168507.

**Author contributions**

A.D.K. conceived the paper and led the writing. All authors contributed to the writing of the manuscript.

**Competing interests**

The authors declare no competing interests.

**Acknowledgements**

A.D.K. and N.J.A. acknowledge support from the Australian Research Council through the Centre of Excellence for the Weather of the 21st Century (CE230100012; A.D.K. and N.J.A.), a Future Fellowship (FT240100306; A.D.K.), and the Special Research Initiative Australian Centre for Excellence in Antarctic Sciences (SR200100008; N.J.A.). A.D.K. and T. Z. acknowledge the Australian Government National Environmental Science Program. E.A.dA. acknowledges funding by the Deutsche Forschungsgemeinschaft (DFG, German Research Foundation) under Germany's Excellence Strategy – EXC 2037 'CLICCS- Climate, Climatic Change, and Society' – Project Number: 390683824

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
