# Peer review of "ESD Ideas: Extended net zero simulations are critical for informed decision making"

_EGUsphere, 2025_

## Author Response (AR1)

**Response to editorial and referee comments**

*We thank the editor for handling our manuscript and providing helpful feedback as well as the reviewers for their constructive comments. Below we detail the changes we've made with point-by-point responses to all comments. Editorial and reviewer comments are in bold text and responses in italics. Line numbers refer to the tracked changes version of the manuscript for ease of reference. There are some slight differences between the changes we proposed and the changes made that arose when making edits, but we believe these are all minor.*

*We have sought to include sufficient discussion of EMICs and relevant studies, as well as further discussion of motivation for analysing regional climates and extremes in extended net zero emissions simulations. As this led to more text being added we have also removed some text which we felt became superfluous to the discussion, specifically the sentences around results from fixed concentration runs using GCMs which we felt could confuse the message of this piece. (see L60-69). We have also added examples of extreme index changes to the Figure as these are more relevant to the use of ESMs for long net zero emissions simulations.*

**Dear Andrew and co-authors,**

**While the reviewers find your work of potential interest, they raise important points that must be addressed before the manuscript can be considered for publication in ESD. In particular, the need for millennial ESM (as opposed to EMIC) simulations and the relevance to decision-making need to be more thoroughly motivated. Earlier literature on millennial ZEC simulations in support of IPCC assessments also needs to be acknowledged (e.g. Plattner et al., J. Clim. 2008; Zickfeld et al., J. Clim, 2013).**

**Best regards,**

**Kirsten**

**References**

**Plattner, G., et al., 2008: Long-Term Climate Commitments Projected with Climate–Carbon Cycle Models. J. Climate, 21, 2721–2751, https://doi.org/10.1175/2007JCLI1905.1.**

**Zickfeld, K., et al., 2013: Long-Term Climate Change Commitment and Reversibility: An EMIC Intercomparison. J. Climate, 26, 5782–5809, https://doi.org/10.1175/JCLI-D-12-00584.1.**

*We thank the editor for their comments and suggested studies to use and cite. We have used both of these studies noting work done using EMICs (L35-40).*

*Reply to Referee 1. We thank the referee for their constructive feedback. Their comments are shown in bold with our responses in italics.*

**Overall evaluation:**

**The paper makes an argument for extending zero emissions simulations to millennial timescales. I strongly agree with the authors that having such simulations would be very useful for our science. However, millennial length simulations of zero emissions do exist for intermediate complexity Earth system models (EMICs), and were in fact part of the original ZECMIP (MacDougall et al, 2020). The authors need to better articulate why**

**simulations with full ESMs are needed at millennial timescales to supplement the results from the EMICs.**

*Thanks. Yes, we previously hadn't articulated well the specific need for long net zero ESM simulations relative to those from EMICs. We added discussion of EMICs, previous analysis using them and have sought to better articulate the merits of using ESMs. (L24-31, 35-40)*

**General Comments:**

**(1) The reason for using ESMs is given at line 70 of the paper "This is particularly problematic for understanding changes in interannual-to- multidecadal climate variability for which changes are only likely to be detectable over long periods. The same is true for multi-year extremes, such as multi-year droughts (Falster et al., 2024), for which limited information may be gained from shorter simulations." EMICs are unable to quantify such metrics, thus justifying the expense of using ESMs. I suggest articulating this point much earlier in the paper.**

*Indeed- we edited the very first paragraph in the main text to highlight specific benefits of long ESM simulations, including analysis of extremes such as drought (L24-26).*

**(2) Early in the paper you should acknowledge the millennial length simulations of zero emissions done with EMICs, summarize what they show then highlight the limitations of such EMIC simulations, thus highlighting the need for longer ESM simulations.**

*Yes, we have added sentences on previous work using EMICs and the general results found across these studies. We have tried to balance providing sufficient detail with not adding too much text. (L24-31, 35-40)*

**(3) It is also important to acknowledge that ESMs often lack processes the EMICs include, which become more important on long timescales. For millennial length simulations processes that are particularly important are: i) dynamic vegetation, ii) ice sheets, iii) permafrost carbon iv) ocean floor carbonate dissolution dynamics. Combined these feedbacks will strongly affect global CO2 concentration, ocean circulation, and regional climates. Also important to note is that many ESMs do not conserve mass and energy to machine precision and thus are not intended for millennial length simulations. Therefore only ESMs with little to no drift in their zero emissions pi-control simulation should be extended to millennial length (hopefully this will be less of a problem for CMIP7 models).**

*We agree these are important points to note. We now mention some benefits of using EMICs (L35-37) and have also added an explicit note on the need to ensure lack of drift in extended ESM simulations (L111-114).*

**Specific comments:**

**Line 18: "Humanity must achieve net zero emissions to slow down climate change" is not really correct. Reducing the rate of emissions should slow down climate change, since warming is roughly proportional to cumulative emissions. To stop global average climate change ZECMIP showed the near-zero emissions are needed. With the ZEC range implying that slightly positive to slightly negative emissions are compatible with zero global temperature change.**

*Yes, we see what you mean and agree this wasn't correct. We edited this from "Humanity must achieve net zero emissions to slow down climate change and to have any hope of meeting the*

*Paris Agreement goals of limiting global warming to well below 2°C." to "Humanity must achieve net zero emissions to have any hope of meeting the Paris Agreement goals of limiting global warming to well below 2°C." (L20).*

**References:**

MacDougall AH, Frölicher TL, Jones CD, Rogelj J, Matthews HD, Zickfeld K, Arora VK, Barrett NJ, Brovkin V, Burger FA, Eby M. Is there warming in the pipeline? A multi-model analysis of the Zero Emissions Commitment from CO 2. Biogeosciences. 2020 Jun 15;17(11):2987-3016.

*Reply to Referee 2. We thank the referee for their constructive feedback. Their comments are shown in bold with our responses in italics.*

**Overall evaluation**

**The authors present an argument for extending ESM simulations under zero emissions to better understand the long term global and regional climate response. Though I agree this would generate useful information, the authors need to be more specific about what they are expecting to gain from model runs of this length, and why ESMs in particular are required. To support the paper's title, the text would also benefit from an example of how millennial-scale ESM runs could impact decision making today.**

*We agree that the motivation for extended runs should have been clearer and specifically why ESMs are needed for this purpose (as opposed to only EMICs; see also reviewer 1's comments and our responses). We have made edits throughout to better distinguish between analysis based on EMICs and why ESMs are also needed as well as specific additional text discussing EMICs and ESMs and their uses (L24-31, 35-40). We have also added some additional discussion on the benefits for decision-making of long ESM runs expanding on the example of droughts (L97-101, 122-124).*

**General comments**

**The authors mention the significance of ESM runs for regional predictions. The proposal would benefit from a short explanation as to how millennial-length ESM simulations would enable this. For example, what are we expecting to gain from these runs vs multi-century simulations, and why are ESMs required over EMICs? Providing examples of which processes are resolved in ESMs but not EMICs, and what impacts these might reveal on millennial timescales would make the argument stronger.**

*Indeed- this wasn't well enough articulated in the initial submission (also noted by Reviewer 1). A key example is for understanding changes in extremes where EMICs can't be used (noted in L16-17 and 24-28). Returning to the drought case, there is a need for long-term planning to determine if augmentation of existing water supply is needed and we extend this discussion (L97-101, 122-124).*

**The case for informing decision making could also benefit from an example. What is it about millennial scale impacts that would change decisions made today?**

*This is in part about highlighting that there are long-term consequences to emissions delay and also about ensuring adequate sample sizes for analysis of rare events. We have expanded the bullet points to highlight this with the drought example (L122-124).*

**The authors need to be careful to differentiate between time, warming and emissions. In Figure 1 they show the change in impacts, largely to the ocean and cryosphere, in a model that stops emissions in the year 2030 and one that stops in 2060. The authors need to clarify whether it is the time it takes to reach net zero that changes these millennial scale outcomes, or the higher cumulative emissions and warming that has resulted during the delay. This also applies to the text in lines 87-90.**

*In reality, it's hard to separate the effects of additional warming and cumulative emissions because they're intrinsically tied, but we have made edits to highlight the complexity of long-term changes under net zero emissions and their relationship to cumulative emissions/warming and time (L80-82).*

**Specific comments**

**Line 14: 'New millennial-length Earth System Model simulations are required to better understand these committed changes and their dependence on delays in reaching net zero emissions.' The authors should specify what we are expecting to change over that time frame compared to the medium term.**

*We have made edits to the Abstract to note the specific benefits of using ESMs to quantify regional changes and changes in extremes, but we have also tried to not add too much additional text to a section that is meant to be very short (L16-17).*

**Line 17: 'Humanity must achieve net zero emissions to slow down climate change'. Reducing emissions will slow the rate of climate change, and net zero emissions may bring us to a point of temperature stabilisation.**

*Thanks. Yes, reviewer 1 also pointed out this error and this has been corrected. We edited this sentence from "Humanity must achieve net zero emissions to slow down climate change and to have any hope of meeting the Paris Agreement goals of limiting global warming to well below 2°C." to "Humanity must achieve net zero emissions to have any hope of meeting the Paris Agreement goals of limiting global warming to well below 2°C." (L20)*

**Lines 19/21: 'regional and local climates will continue to evolve for many centuries. There is surprisingly little known about these longer timescale climate changes despite their policy relevance'. An example of a regional change would be helpful here. Global climate may also change over these timeframes, bearing in mind the uncertainty in the ZEC assessment.**

*Yes, we agree. We use the example of substantial warming in many Southern Hemisphere land areas found with the 1000-year ACCESS-ESM-1.5 simulations. (L23-24)*

**Figure 1: small suggestion to change the colour scheme to avoid the automatic association between red and warming e.g. Arctic sea ice extent in March is actually increased for a net zero that's imposed later, but is coloured red. It would also be useful to add in the figure caption what the emissions scenario was for these runs.**

*Yes, we see what you mean. We have made changes to the Figure colour scheme to avoid this issue by using green shading. We also provide more detail of the emissions pathway in the*

*caption (L89-90). Both runs branch from SSP5-8.5 with zero emissions imposed from 2030 onwards or 2060 onwards.*

**In Figure 1, it would also be useful to include the overall changes as well as the difference between the two time periods. For example, I suspect the very small reduction in Arctic sea ice for the 2060 run does not reflect less sea ice loss overall, and is instead the result of front-loading the impacts into the first 300 years in the 2060 scenario.**

*The Arctic sea ice extent changes are minimal in the 2060 run mainly due to summer sea ice loss during the 21$^{st}$ century in SSP5-8.5 and a lack of recovery in the net zero simulations (see Figure 3f in King et al. 2024). Again, we can see the benefit in showing previous or overall changes for context, but equally we don't want to make the Figure too busy. We have noted this in the caption instead though (L91-92).*

**Line 76 is an important point and would benefit from coming much earlier, possibly in the abstract. 'The lack of extended net zero simulations risks understating the long-lasting impacts of increased global temperatures relative to pre-industrial levels'.**

*We agree. We made an edit to the first paragraph (L30-31) in the main text to note this.*

**Line 78: The authors need to explain why extending these runs will offer the same or better utility than the alternative of extending the 1pct and bell emissions pathways to fall in line with the ZECMIP experiments.**

*Yes, we agree the ZECMIP runs could also be extended. Given the setup of ZECMIP (with branching from a concentration-driven run) and the shift towards emissions-driven model setups we proposed extensions from either flat10MIP or TipMIP with the view that these setups may be more likely to continue in future rounds of CMIP. We made an edit to note this (L109-111).*

---

## Editor Decision (ED1)

**Editor comments**

l. 20 achieve net zero emissions -> need to indicate timing (2nd half of 21st century).

l. 70 see also Gillet et al., 2011 (https://www.nature.com/articles/ngeo1047).

l. 119: Does delay in achieving net zero imply different amounts of cumulative emissions and therefore peak warming levels? Please clarify.

---

## Author Response (AR2)

**Response to editorial and referee comments**

*We thank the editor for their feedback as well as the reviewers for looking over the manuscript again. We provide responses to the three editorial comments below. Our responses are italicised and line numbers refer to the tracked changes version of the document.*

**l. 20 achieve net zero emissions -> need to indicate timing (2nd half of 21st century).**

*We have added a mention of timing as suggested (L19).*

**l. 70 see also Gillet et al., 2011 (https://www.nature.com/articles/ngeo1047).**

*Thanks for the relevant paper. We have added a reference to it (L71).*

**l. 119: Does delay in achieving net zero imply different amounts of cumulative emissions and therefore peak warming levels? Please clarify.**

*We think this comment is meant to be for L109 but please let us know if this is incorrect.*

*Yes, when we refer to delay we also mean higher cumulative emissions and peak global warming. We have edited this sentence to note this (L109-111).*